# Assessments of Cloud Liquid Water and Total Precipitable Water Derived from FY-3E MWTS-III and NOAA-20 ATMS

**Changjiao Dong** [1,2,3], **Fuzhong Weng** [2,3,*] **and Jun Yang** [2,3]

1   School of Atmospheric Physics, Nanjing University of Information Science and Technology, Nanjing 210044, China; dongcj1997@gmail.com
2   CMA Earth System Modeling and Prediction Centre (CEMC), China Meteorological Administration, Beijing 100081, China; yangjun@cma.gov.cn
3   State Key Laboratory of Severe Weather, Chinese Academy of Meteorological Sciences, Beijing 100081, China
*   Correspondence: wengfz@cma.gov.cn

**Abstract:** Cloud liquid water (CLW) and total precipitable water (TPW) are two important parameters for weather and climate applications. Typically, microwave temperature sounding instruments onboard satellites are designed with two low-frequency channels at 23.8 and 31.4 GHz and can be used for retrieving CLW and TPW over global oceans. Since MWTS-III polarization at above two frequencies is uncertain, we must first determine their polarization involved in retrieval algorithms. Through radiative transfer simulation, we found that uses of the quasi-horizontal polarization for MWTS-III can produce smaller biases between observations and simulations and the scan-angle dependence of the biases is also in a general frown pattern, which is similar to ATMS pitch-maneuver observations. After the characterization of MWTS-III polarization, CLW and TPW are derived from Microwave Temperature Sounder (MWTS-III) and are compared with those from ATMS. It is found that CLW and TPW derived from two instruments exhibit a high consistency in terms of their spatial distributions and magnitudes.

**Keywords:** cloud liquid water; total precipitable water; MWTS; ATMS

## 1. Introduction

Atmospheric water in vapor and liquid phases affects global weather and climate through its radiation and hydrological processes, and its distribution and evolution also influence the climate change [1–3]. Total precipitable water is defined as vertically integrated water vapor content over an area, and the knowledge of its magnitude in the atmosphere can significantly improve weather forecasts of precipitation [4]. The interaction between radiation and water vapor can significantly influence the feedback mechanism in the climate system [2,3,5].

Since the radiosonde observations measure atmospheric humidity, they are ideal for deriving the integrated water vapor content in the atmospheric column [6]. Ground-based microwave radiometry measurements are also used to determine cloud liquid water and precipitable water simultaneously [7–9]. However, these techniques are mostly limited over global land and are available with limited temporal information [10]. Thus, the observations from a variety of satellite and ground-based instruments are also used to retrieve cloud liquid water and total precipitable water. Compared with visible and infrared observations, microwave observations at K and Ka bands respond to atmospheric thermal radiation and are directly linked to the emission from cloud and water vapor. Furthermore, spaceborne microwave observations are often used to estimate the cloud liquid water and water vapor over oceans, where surface emitted radiation is relatively lower and can be reliably estimated [11,12].

The majority of the cloud liquid water retrieval algorithms were designed for conically scanning microwave radiometers, such as SMMR, SCAMS, TMI, GMI, AMSR, AMSR2,

and SSM/I. The pioneer work was conducted in the 1970s by Grody to analyze the data from a scanning microwave spectrometer (SCAMS) and to determine the relationship between the brightness temperatures at 21 and 31 GHz and cloud liquid water [13]. As the passive microwave technology evolved, more and more researchers have developed their algorithms for calculating the distribution of total precipitable water (TPW) and cloud liquid water (CLW) over global oceans [14–25]. The majority of earlier algorithms are aimed at applications over the oceans and some are also developed for total precipitable water over land [10,26]. Cloud liquid water can also be derived over land from visible wavelengths [22,27–29]. For simultaneously retrieving TPW and CLW, the algorithms typically require the measurements at two frequencies, which orthogonally respond to the radiation from clouds and water vapor. In 1994, Weng and Grody first combined three pairs of retrieval algorithms, with each suitable for measuring a specific range of cloud water. Thus, this composite algorithm can retrieve the cloud liquid water content under a variety of atmospheric conditions [23]. In 2017, the observations from the Fengyun-3 satellite microwave radiation imager (MWRI) were used to retrieve the cloud liquid water path with a largest dynamic range [30]. For a cross-track scanning microwave radiometer, the data from the advanced microwave sounding unit (AMSU) onboard NOAA-15 satellite were first used to retrieve the cloud liquid water. AMSU has 15 channels with a frequency ranging from 23 to 89 GHz. Temperature and humidity profiles can be derived in combining AMSU-A with the microwave humidity sounder (AMSU-B/MHS). Grody et al. (2001) [31] initially developed a statistical algorithm for AMSU-A cloud liquid water algorithm, following the approach similar to SSM/I algorithms, and then Weng et al. (2003) [25] designed a full physical algorithm for simultaneously deriving both cloud liquid water and water vapor.

In Weng's physical retrieval algorithms, the coefficients are parameterized as a function of surface emissivity and temperatures. Thus, the algorithms require the observed brightness temperature and corresponding surface parameters as inputs. For NOAA microwave sounding instruments, through the study of ATMS instrument performance, data quality, and other factors, ATMS is found to be superior to AMSU-A/MHS in several aspects [32,33]: ATMS has an improved spatial resolution for most temperature sounding channels. Its gap between two consecutive orbits is very small [34] and, thus, is more suitable for monitoring clouds and precipitation over global oceans. With a newly launched FY-3E early-morning satellite, the new-generation microwave temperature sounder (MWTS-III) offers even more temperature sounding channels than AMSU and ATMS, and it has a wider scan swath (~3100 km). MWTS-III has 17 channels, 15 of which are inherited from the original MWTS-II and ATMS/AMSU-A.

In this study, the CLW and TPW algorithms previously developed for AMSU-A are utilized for MWTS-III and ATMS. In Section 2, ATMS and MWTS-III instruments are briefly discussed and some specific instrument features of MWTS-III affecting the CLW and TPW retrievals are analyzed. Section 3 provides an overview of CLW and TPW algorithms. In Section 4, the retrieved CLW and TPW from MWTS-III are compared with those from ATMS for a consensus assessment. Section 5 summarizes the conclusions from this study.

## 2. Instrument Characteristics

### 2.1. ATMS vs. MWTS-III

The Advanced Technology Microwave Sounder (ATMS) was first carried onboard the National Polar-orbiting Partnership (Suomi NPP) satellite of the United States' new-generation Joint Polar Satellite System (JPSS). It combines the advanced microwave sounding unit-A (AMSU-A) and AMSU-B/microwave humidity sounder (MHS) that can probe both atmospheric temperature and humidity. ATMS has 22 channels in total, with the first, second, and sixteenth channels being quasi-vertical polarization and the remaining channels being quasi-horizontal polarization. The first 15 ATMS channels are primarily used to retrieve temperature profiles and are located within a frequency range of 23.8–57.3 GHz. With the exception of the first two channels, 13 channels are located near the 50–60 GHz oxygen absorption band and have a temperature weighting function peaking from the

surface to 50 km above [35]. ATMS's last seven high-frequency channels are primarily designed for humidity profiles. Specific channel parameters are shown in Table 1. Notice ATMS also has a scan swath of 2300 km covered through 96 scan positions. For channel 1 and 2, the resolution near satellite nadir is 75 km and, for channels 3–16, the resolution is about 30 km [32].

**Table 1.** ATMS instrument characteristics.

| Channel Number | Center Frequency (GHz) | Bandwidth (MHz) | NEΔT (K) |
|:---:|:---:|:---:|:---:|
| 1 | 23.8 | 270 | 0.7 |
| 2 | 31.4 | 180 | 0.8 |
| 3 | 50.3 | 180 | 0.9 |
| 4 | 51.76 | 400 | 0.7 |
| 5 | 52.8 | 400 | 0.7 |
| 6 | $53.596 \pm 0.115$ | $2 \times 170$ | 0.7 |
| 7 | 54.4 | 400 | 0.7 |
| 8 | 54.94 | 400 | 0.7 |
| 9 | 55.5 | 330 | 0.7 |
| 10 | 57.29(fo) | 330 | 0.75 |
| 11 | $fo \pm 0.217$ | $2 \times 78$ | 1.2 |
| 12 | $fo \pm 0.3222 \pm 0.048$ | $4 \times 36$ | 1.2 |
| 13 | $fo \pm 0.3222 \pm 0.022$ | $4 \times 16$ | 1.5 |
| 14 | $fo \pm 0.3222 \pm 0.010$ | $4 \times 8$ | 2.4 |
| 15 | $fo \pm 0.3222 \pm 0.0045$ | $4 \times 3$ | 3.6 |
| 16 | 88.2 | 2000 | 0.5 |
| 17 | 165.5 | 3000 | 0.6 |
| 18 | $183.31 \pm 7.0$ | 2000 | 0.8 |
| 19 | $183.31 \pm 4.5$ | 2000 | 0.8 |
| 20 | $183.31 \pm 3.0$ | 1000 | 0.8 |
| 21 | $183.31 \pm 1.8$ | 1000 | 0.8 |
| 22 | $183.31 \pm 1.0$ | 500 | 0.9 |

The Fengyun-3E (FY-3E) was successfully launched on the Long March-4C at the Jiuquan Satellite Launch Center on 5 July 2021. Fengyun 3E is the world's first civil early-morning orbit meteorological satellite and will constellate with the FY-3C and FY-3D satellites to provide the data coverage from the early-morning, mid-morning, to afternoon local cross-time. A three-orbit constellation can also be achieved from FY-3E, METOP-C, and NOAA-20 satellites as part of the global observing system planned by the World Meteorological Organization (WMO). This constellation can provide the global atmospheric sounding data for numerical weather prediction (NWP) applications with a four-hour refresh rate. It has the potential to significantly improve the accuracy and effectiveness of global NWP, which is critical to the advancement of the global Earth observation system [36]. FY-3E carries 13 payloads onboard, 3 of which are newly developed, 7 of which are upgraded and improved, and 1 is inherited. Only cloud water products retrieved from ATMS and MWTS-III data are compared in this study.

Compared with MWTS-II, MWTS-III is designed with two new window channels with a center frequency of 23.8 GHz and 31.4 GHz (see Table 2). These channels can be used to detect precipitation, surface emissivity, liquid water, and water vapor [25]. The additional channels at $53.246 \pm 0.08$ and $53.948 \pm 0.081$ GHz can also improve atmospheric temperature structure near the lower troposphere and are not part of ATMS.

MWTS-III has a scan angle up to $53.35°$ from the nadir, which corresponds to a local zenith angle of 70 degree, whereas ATMS has a scan angle up to $52.725°$ and a local zenith angle of $63°$. With a larger range of scan angles, MWTS-III achieves a total of field of views (FOV) of 98 within each scan line and has a scan swath of 3100 km, about 400 km broader than ATMS. Thus, MWTS-III is capable of observing the global atmosphere twice a day. The two instruments also differ in certain channel parameters, such as their polarization.

To illustrate their scan patterns, we also plot the FOV distributions of adjacent scanlines for the first two channels as shown in Figure 1. It is apparent that the scan coverage of MWTS-III is much wider than that of ATMS.

**Table 2.** MWTS-III instrument characteristics.

| Channel Number | Center Frequency (GHz) | Bandwidth (MHz) | NEΔT (K) |
|:---:|:---:|:---:|:---:|
| 1 | 23.8 | 270 | 0.30 |
| 2 | 31.4 | 180 | 0.35 |
| 3 | 50.30 | 180 | 0.35 |
| 4 | 51.76 | 400 | 0.30 |
| 5 | 52.8 | 400 | 0.30 |
| 6 | 53.246 ± 0.08 | 2 × 140 | 0.35 |
| 7 | 53.596 ± 0.115 | 2 × 170 | 0.30 |
| 8 | 53.948 ± 0.081 | 2 × 142 | 0.35 |
| 9 | 54.40 | 400 | 0.30 |
| 10 | 54.94 | 400 | 0.30 |
| 11 | 55.50 | 330 | 0.30 |
| 12 | 57.29(fo) | 330 | 0.60 |
| 13 | fo ± 0.217 | 2 × 78 | 0.70 |
| 14 | fo ± 0.322 ± 0.048 | 4 × 36 | 0.80 |
| 15 | fo ± 0.322 ± 0.022 | 4 × 16 | 1.00 |
| 16 | fo ± 0.322 ± 0.010 | 4 × 8 | 1.20 |
| 17 | fo ± 0.322 ± 0.0045 | 4 × 3 | 2.10 |

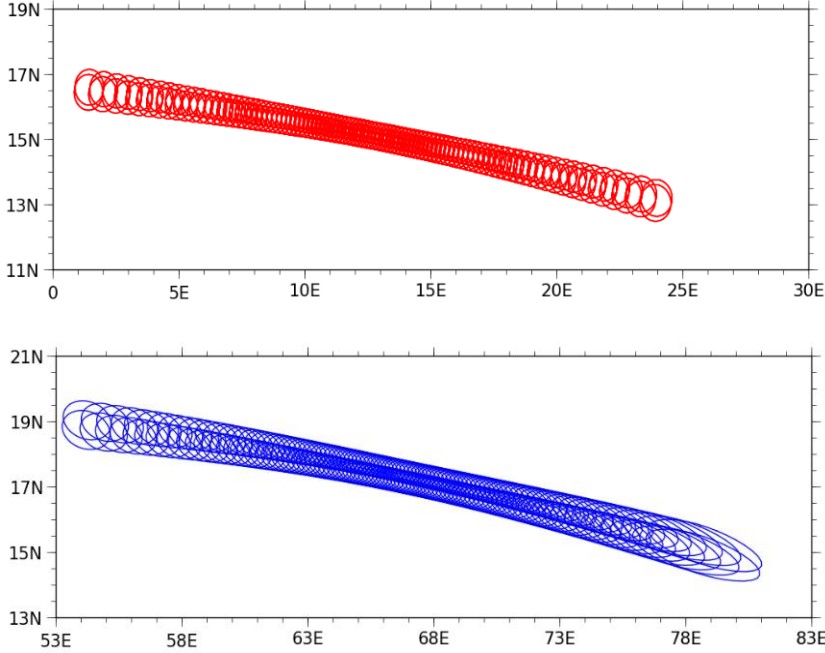

**Figure 1.** ATMS (**top**) and MWTS (**bottom**) field of view (FOV) dimensions for two adjacent scan lines with 5.2° beam width.

*2.2. Determination of MWTS-III Channel Polarization*

Most instruments onboard FY-3E satellite are still in their intensive calibration and validation stage. For MWTS-III, its polarization is not specified clearly during the prelaunch phase; thus, its quantitative applications might be affected if its polarization is uncertain. Therefore, a method is developed for characterizing MWTS-III polarization.

Weng and Yang (2013) [37] compared the antenna temperature of the ATMS quasi-vertical (*qv*) and quasi-horizontal (*qh*) polarization channels from its pitch maneuver cold

space observations. For a *qv* polarization channel, cold space antenna temperatures exhibit a "smile" shape with a scan angle, whereas, for the *qh* polarization channel, it shows a "frown" shape. Thus, this dependence can be explained by an emitting antenna. For *qv* channels, the actual radiance reaching the radiometer receiver is:

$$T_{qv}^c = T_{qv} + \varepsilon_h(T_r - T_h) + [\varepsilon_v(T_r - T_v) - \varepsilon_h(T_r - T_h)]\sin^2\theta - \frac{T_3}{2}(1 - \varepsilon_h)^{3/2}\sin 2\theta \quad (1)$$

For *qh* polarization:

$$T_{qh}^c = T_{qh} + \varepsilon_h(T_r - T_h) + [\varepsilon_v(T_r - T_v) - \varepsilon_h(T_r - T_h)]\cos^2\theta + \frac{T_3}{2}(1 - \varepsilon_h)^{3/2}\sin 2\theta \quad (2)$$

where $T_{qv}$ and $T_{qh}$ are the brightness temperatures at *qv* and *qh*, which are further related to the brightness temperatures at pure vertical and horizontal polarization.

$$T_{qv} = T_h\sin^2\theta + T_v\cos^2\theta \quad (3)$$

$$T_{qh} = T_h\cos^2\theta + T_v\sin^2\theta \quad (4)$$

where $\varepsilon_v$ and $\varepsilon_h$ are the reflector emissivity at the vertical and horizontal polarization, respectively, $T_3$ is the third Stokes component of the scene, $T_r$ is the temperature of the reflector, and $\theta$ is the scan angle. Note that $\varepsilon_v = 2\varepsilon_h - \varepsilon_h^2$ at an incident angle of 45 degrees to the reflector normal [38].

For ATMS, this model can well explain the scan-angle dependence of the difference between observed and simulated brightness temperatures under various oceanic conditions [38].

In this investigation, the brightness temperatures of ATMS and MWTS-III at channels 1 and 2 are also simulated over the oceans. The advanced radiative transfer modeling system (ARMS) is used to carry out the simulations. The ARMS can simulate many satellite instruments, including China's FengYun satellite [39]. The ERA5 reanalysis data of atmospheric profiles (such as sea surface wind speed, temperature, humidity, etc.) are used as inputs to ARMS.

In ARMS, two versions of ocean emissivity models (FASTEM5/6) are also tested for better understanding of the scan-angle dependent biases. The simulations from clear-sky FOVs are used for bias assessments. The clear-sky FOVs are determined by FY-4A AGRI cloud masks. As shown in Figure 2, ATMS biases at 23.8 GHz and 31.4 GHz are slightly affected by the surface emissivity models. At a large scan angle, it is clear that the bias from FASTEM6 is smaller than that from FASTEM5. Overall, the scan-angle dependent bias corresponds to the features at the quasi-vertical polarization [38].

Since the polarization mode of the MWTS-III channels is uncertain, brightness temperatures at 23.8 GHz and 31.4 GHz are simulated with both *qv* and *qh* modes. ARMS surface emissivity is configured with FASTEM6. Figure 3a,b show the observations and simulations and their difference (bias) with scan angle for the quasi-vertical polarization. The symbol (O) represents the observed brightness temperature, B represents the simulated brightness temperature, and O−B is the bias between the observed and simulated brightness temperatures. Figure 3c,d show the results corresponding to the quasi-horizontal polarization. It is clearly seen that the biases from the quasi-horizontal polarization are significantly smaller than those from the quasi-vertical polarization. Thus, in our following studies, the polarization of the first two channels of MWTS-III is assumed as quasi-horizontal polarization.

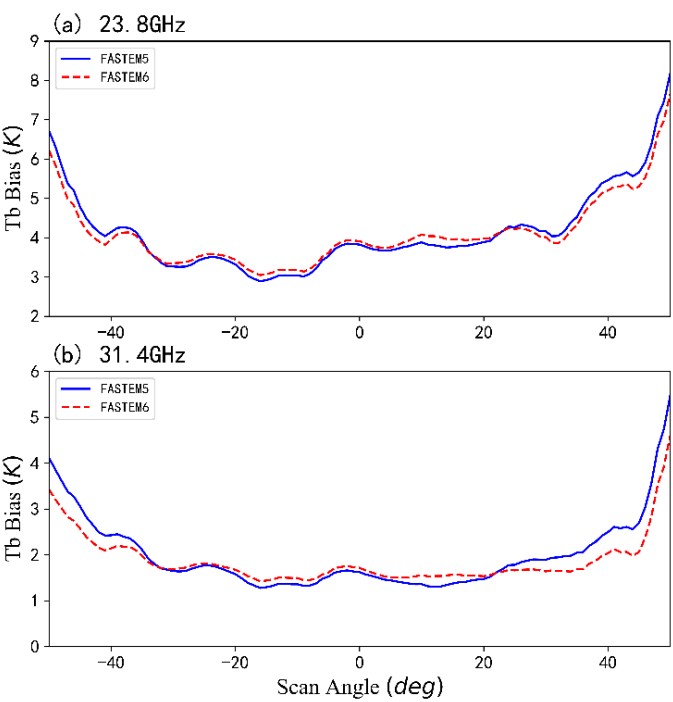

**Figure 2.** Mean biases (O−B, unit: K) of simulated brightness temperatures from observed temperatures versus scan angle (Abscissa, unit: degree) under clear atmosphere over oceans at (**a**) 23.8 GHz and (**b**) 31.4 GHz.

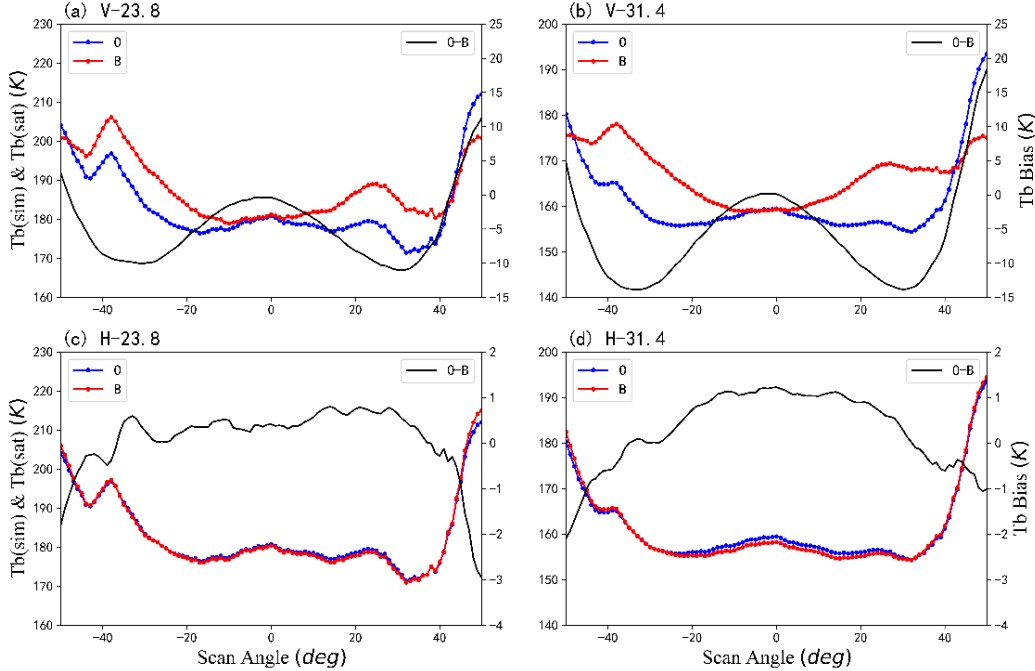

**Figure 3.** Mean bias (O−B, unit: K) of observed (O, unit: K) and simulated (B, unit: K) brightness temperatures under clear atmospheres over oceans at (**a**) 23.8 GHz with *qv*-pol; (**b**) 31.4 GHz with *qv*-pol; (**c**) 23.8 GHz with *qh*-pol; (**d**) 31.4 GHz with *qh*-pol.

## 3. *CLW* and *TPW* Algorithms

For a scattering-free atmosphere, the radiative transfer equation can be simplified as:

$$T_b = T_s[1 - (1 - \varepsilon)Y^2] - \Delta T(1 - Y)[1 + (1 - \varepsilon)Y] \tag{5}$$

where $\Delta T = T_s - T_m$, $T_s$ is the surface temperature, $T_m$ is the atmospheric temperature, and $\varepsilon$ is the surface emissivity, and $Y$ is the atmospheric transmittance and is further related to the optical depth and local zenith viewing angle. Equation (5) is a foundation for remote sensing of cloud liquid water and total precipitable water, as discussed in many previous studies.

If the atmosphere is assumed as isothermal ($\Delta T = 0$), Equation (5) is simplified as:

$$T_b = T_s[1 - (1 - \varepsilon)Y^2] \tag{6}$$

From Equation (6), Weng et al., 2003 derived more general expressions for *CLW* and *TPW* retrievals as follows:

$$CLW = a_0\mu[\ln(T_s - T_{b31}) - a_1\ln(T_s - T_{b23}) - a_2] \tag{7}$$

$$TPW = b_0\mu[\ln(T_s - T_{b31}) - b_1\ln(T_s - T_{b23}) - b_2] \tag{8}$$

where $\mu$ is cosine of the local zenith angle, $T_{b23}$ and $T_{b31}$ are the brightness temperature of channels 23.8 and 31.4 GHz, respectively, $T_s$ is the surface temperature, and the coefficients $a_0$, $a_1$, $a_2$, $b_0$, $b_1$, and $b_2$ are defined as further linked to the surface parameters, such as sea surface wind and temperature [25]. Grody et al. (2001) set $T_s$ as a constant of 285 K and all the coefficient fixed values, and thus derived a statistical algorithm for easy applications, but the algorithms do not warrant the accuracy [31].

Note that Equation (6) is derived under a scattering-free and isothermal atmospheric condition and, thus, it has a bias to the full radiative transfer model. Following our previous approaches used for AMSU-A [25], brightness temperatures from ATMS and MWTS-III channel 1 and 2 at 23.8 and 31.4 GHz must be corrected to the values computed from Equation (6) prior to their uses in Equations (7) and (8). In this study, we derive this expression for correcting the measurements to simulations as:

$$\Delta T_b = A_0 \exp\left\{-\frac{1}{2}[(\theta_s - A_1)/A_2]^2\right\} + A_3 + A_4\theta_s + A_5\theta_s^2 \tag{9}$$

where $\theta_s$ is the scan angle and the coefficients are derived separately for satellite ascending and descending nodes and are given in Tables 3 and 4.

**Table 3.** Coefficients used to correct NOAA-20 ATMS cross-track asymmetry.

| Orbit Node | Frequency (GHz) | $A_0$ | $A_1$ | $A_2$ | $A_3$ | $A_4$ | $A_5$ |
|---|---|---|---|---|---|---|---|
| ascending | 23.8 | 7.56086 | 0.599034 | 31.9538 | −5.66606 | −0.0024 | 0.002557 |
| | 31.4 | 0.498883 | 14.7335 | −3.69799 | 0.280111 | −0.01215 | 0.000466 |
| descending | 23.8 | 0.588579 | 8.67488 | 12.2642 | 0.562151 | −0.00766 | 0.00058 |
| | 31.4 | 1.12823 | 8.7327 | 18.2881 | −0.58358 | −0.0138 | 0.000838 |

**Table 4.** Coefficients used to correct FY-3E MWTS-III cross-track asymmetry.

| Orbit Node | Frequency (GHz) | $A_0$ | $A_1$ | $A_2$ | $A_3$ | $A_4$ | $A_5$ |
|---|---|---|---|---|---|---|---|
| ascending | 23.8 | −36111.1 | −2.11963 | 315.961 | 36108.3 | −0.760975 | −0.179592 |
| | 31.4 | −15.7695 | −7.13451 | 49.5332 | 14.5176 | −0.036707 | −0.002507 |
| descending | 23.8 | −67476.8 | −3.62786 | 375.999 | 67471.6 | −1.72343 | −0.237488 |
| | 31.4 | −0.718871 | −13.2727 | 17.7433 | −0.171598 | −0.005638 | −0.000172 |

Figure 4 shows the mean bias between observations and simulations (O-B) verse scan angle for ATMS, where B is simulated from Equation (6). In the simulations, ATMS

and MWTS-III observations from July 12 to July 31, 2021 within the latitude range of 55°S–55°N are first collocated with ECMWF reanalysis version 5 (ERA5) data and the ERA5 atmospheric and surface parameters are used for calculation in Equation (6). It is apparent that initial O-B has a strong angle dependent. After applying Equation (9), the O-B becomes less variable uniform across the scan angle. Likewise, Figure 5 depicts MWTS-III O-B at channels 1 and 2. It is very interesting to see brightness temperatures at one side of the scan edge always exhibit larger biases than those at the other side. Indeed, this asymmetry is also shown in ARMS simulation. Thus, this asymmetric bias must be corrected in order to produce better *CLW* and *TPW* products across all scan positions.

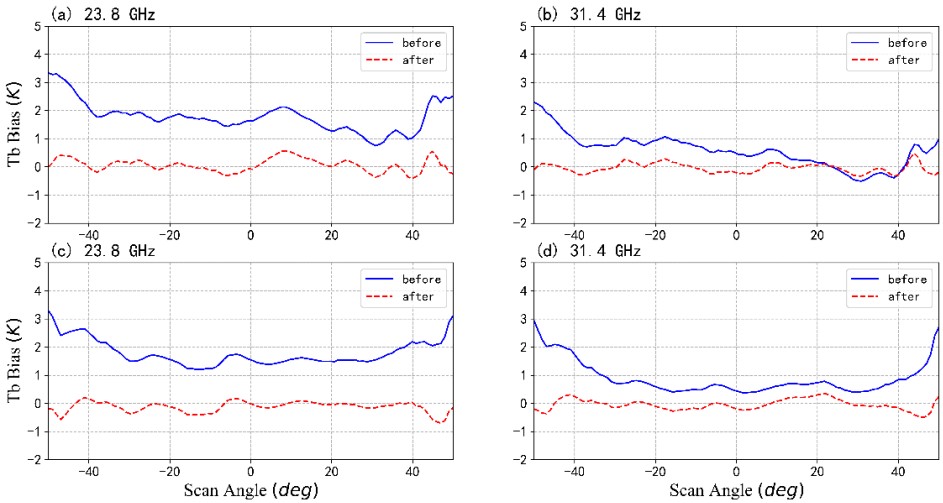

**Figure 4.** ATMS $T_b$ bias before and after correction for asymmetry of different channel frequencies at (**a**,**b**) ascending; (**c**,**d**) descending.

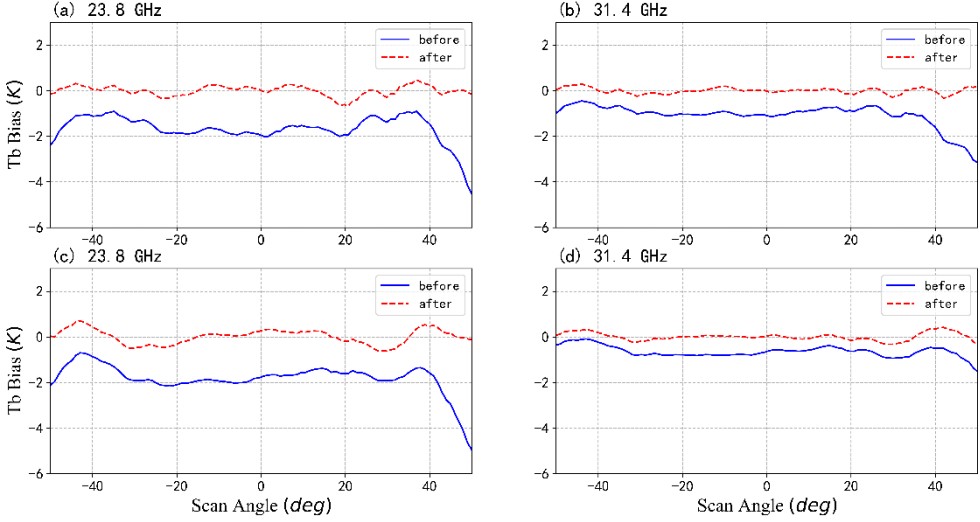

**Figure 5.** MWTS-III $T_b$ bias before and after correction for asymmetry of different channel frequencies at (**a**,**b**) ascending; (**c**,**d**) descending.

## 4. Retrieval Results

Figures 6 and 7 show ATMS *CLW* and *TPW* over global oceans on September 13, 2021, respectively. Notice European Center for Medium-range Weather Forecast (ECWMF) Reanalysis (version 5)-ERA5 analysis field [17] is shown in the middle panel of figure for a sanity check. In principle, both parameters should be verified against the in situ observations. However, over the vast global oceans, radiosonde measurements on atmospheric

water vapor are limited to certain island stations and cloud liquid water measurements are rarely present. Thus, ERA5 products are used as an alternative for a comparison. In fact, ERA5 data are generated from ECMWF global models that have assimilated the most of available data from satellites and in situ observations, and it is recognized as a high-quality product and can be used for satellite product validation [4] from radiosondes.

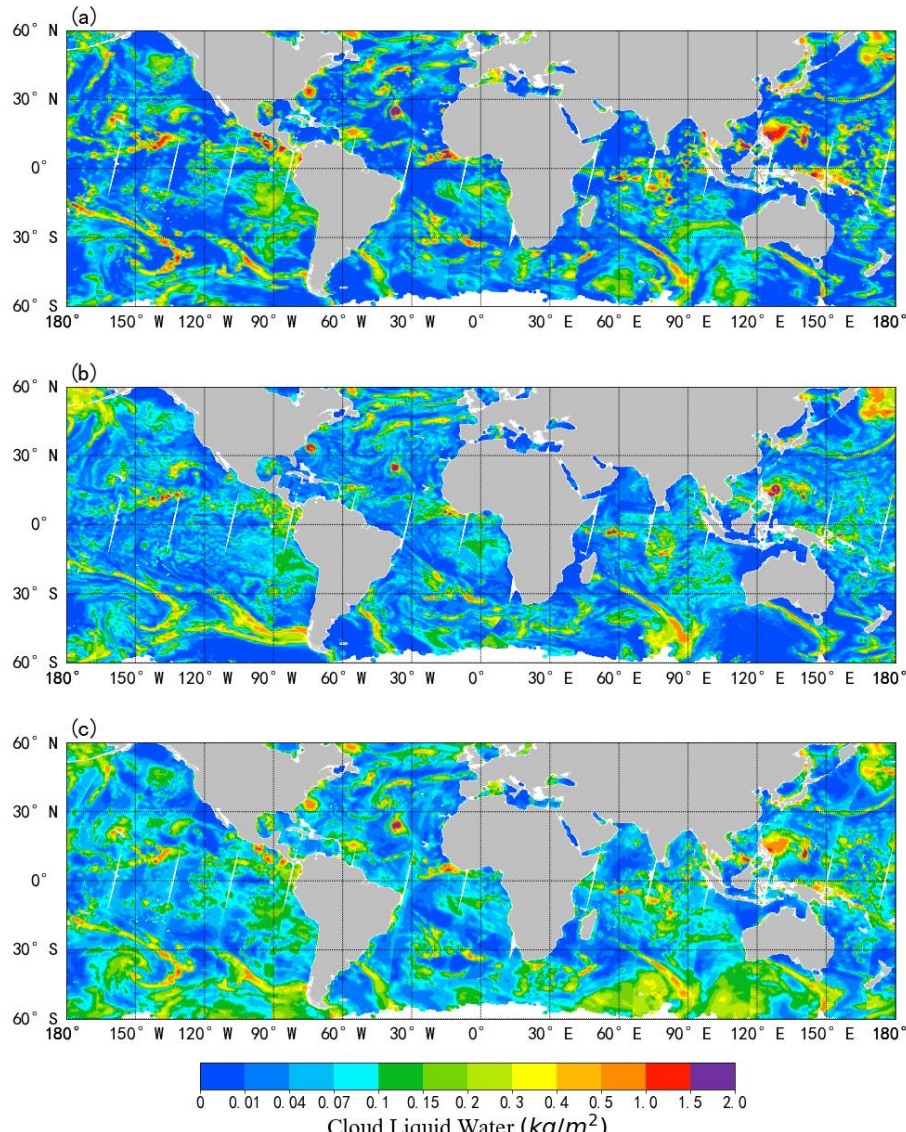

**Figure 6.** *CLW* retrieved from ATMS over global oceans on 13 September 2018 using (**a**) physical algorithm; (**b**) ERA5 reanalysis data; and (**c**) statistical algorithm.

The process for generating the coefficients is different between the physical algorithm and the statistical algorithm. The statistical method takes the representative radiosonde profile as inputs to the radiative transfer model, and the brightness temperatures at the top of atmosphere are then simulated. The simulated data are used to generate the coefficients in the statistical algorithm. The physical algorithm is based on the emission-based radiative transfer equation, which links the brightness temperatures to physical parameters, such as sea surface temperature, emissivity, and atmospheric transmittance. Overall, *CLW* and *TPW* retrieved by both statistical and physical algorithms exhibit a global consistency with the ERA5 reanalysis data. The *CLW* magnitude retrieved by the statistical algorithm in the middle and high latitudes is generally higher than that from the physical algorithm and reanalysis data, as shown in Figure 6. This could be due to uses of constant SST

of 285 K in Grody's algorithm [31]. As shown in Figure 6, the physical algorithm also produces a clear separation of cloudy from clear scenes in the tropics. Moreover, both *CLW* and *TPW* from the physical algorithm are closer to the reanalysis data. In the case of precipitable water, the ERA5 data are produced from assimilation of satellite-borne infrared, microwave observations and many other conventional data. In this study, *CLW* and *TPW* are retrieved simultaneously by selecting two channels, resulting in the same product quality. Specifically, three precipitable products are shown through scatter diagrams, as shown in Figure 8. Comparing Figure 8a,b, it can be seen that the *TPW* retrieved by both the physical and statistical algorithms has a strong correlation with the ERA5 reanalysis data. When various parameters are compared, the physical approach outperforms the statistical algorithm with a small margin.

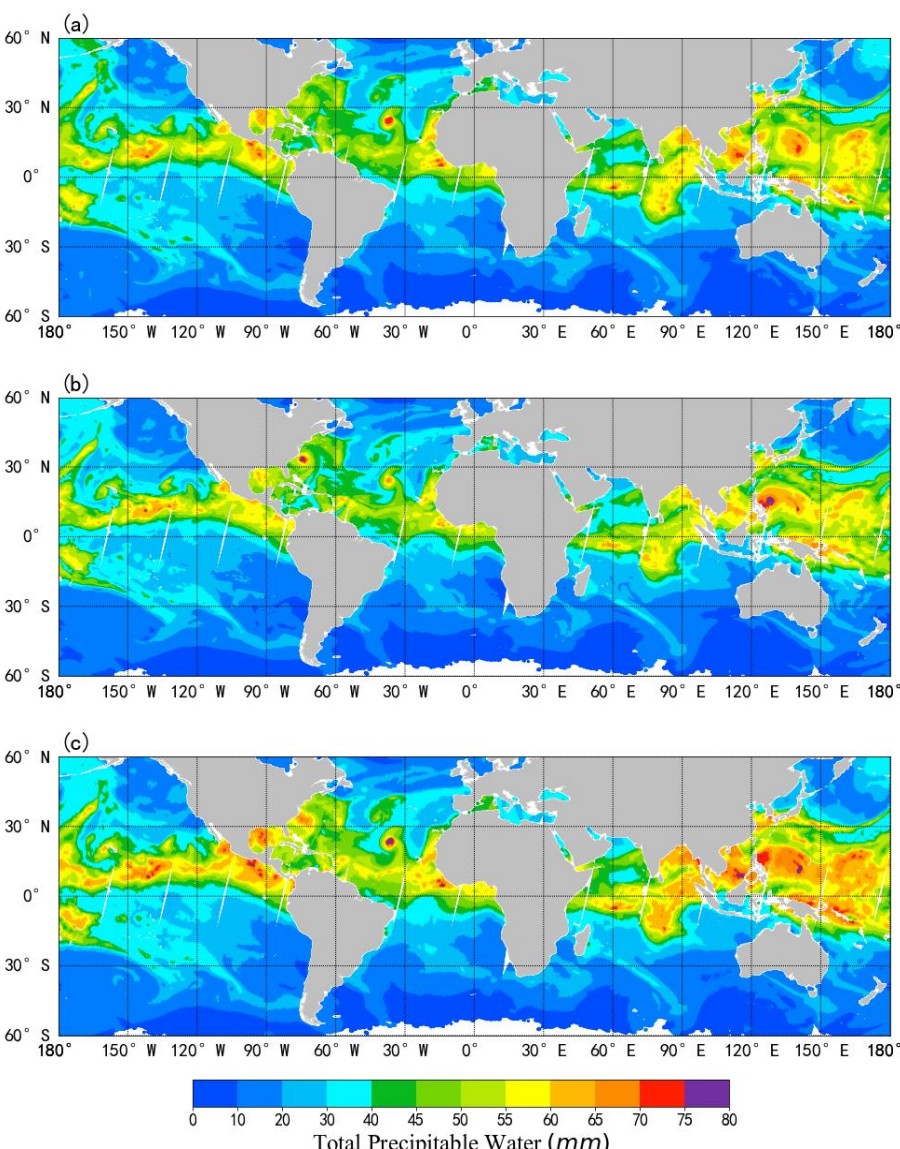

**Figure 7.** *TPW* retrieved from ATMS over global oceans on 13 September 2018 using (**a**) physical algorithm; (**b**) ERA5 reanalysis data; (**c**) statistical algorithm.

For a specific region, Figures 9 and 10 zoom in on more details on *CLW* and *TPW* at ascending and descending orbits, respectively. For tropical storm regions, *CLW* retrieved by the physical algorithm is higher than that from the statistical algorithm. The statistical algorithm produces more clouds and a higher amount of *TPW*. The physical algorithm

defines the cloudy areas more consistently with VIIRS cloudy areas, as shown in Figure 11. In general, cloud liquid water greater than 0.2 mm is best retrieved using frequencies less than 37 GHz, and a higher frequency is required for lower *CLW*. As an independent verification, the VIIRS visible band cloud image displays a noticeable clear sky area between the two cloud clusters near 40°N and 160°E, yet Figure 9a shows a minor amount of *CLW* in this area, which is not reasonable. In addition, the result of 9b correlates to the cloud image (Figure 11) in the same region.

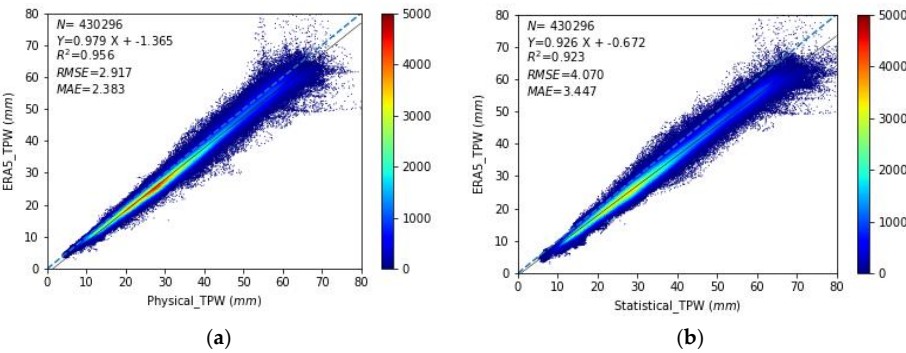

**Figure 8.** Scatter plot of *TPW* over oceans from (**a**) physical and (**b**) statistical descending orbit data collocated with ERA5 *TPW* in 2018.

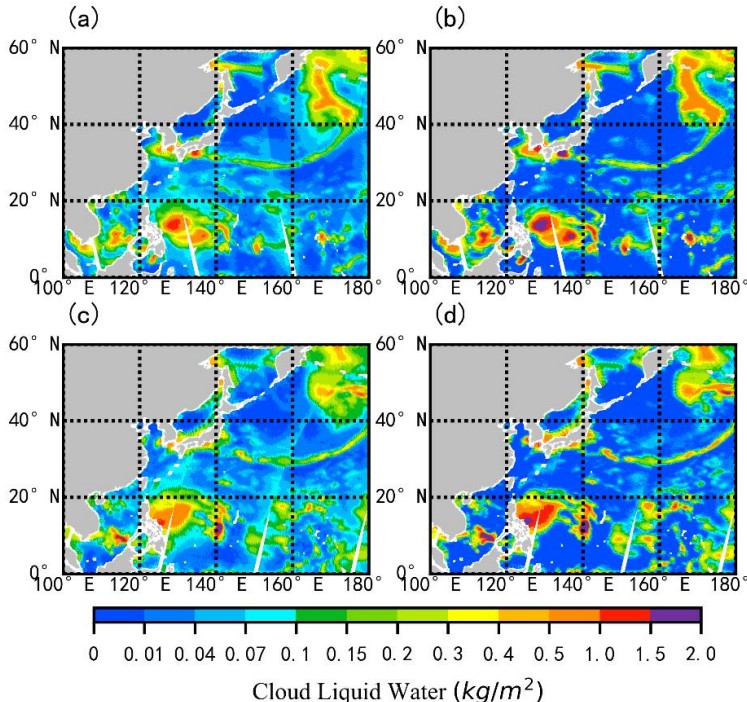

**Figure 9.** *CLW* in the northwest Pacific on 13 September 2018 retrieved from ATMS by using the (**a**) statistical algorithm and (**b**) the physical algorithms at the ascending orbit; (**c**,**d**) those from the descending orbit.

ATMS and MWTS-III ascending orbit observation data from the same day (1 August 2021) and physical algorithms were used to retrieve *CLW* and *TPW*, respectively. However, the observation times of the two satellites carrying the instrument differ (one for the afternoon and one for the early morning). As a result, matching the two instruments in time and space is impossible. We match the ERA5 reanalysis product with the MWTS-III in space and time, and analyze the spatial structure of *CLW* and *TPW* over the global ocean

for these three outcomes, because the ERA5 reanalysis data are hour-by-hour global and their products are relatively reliable.

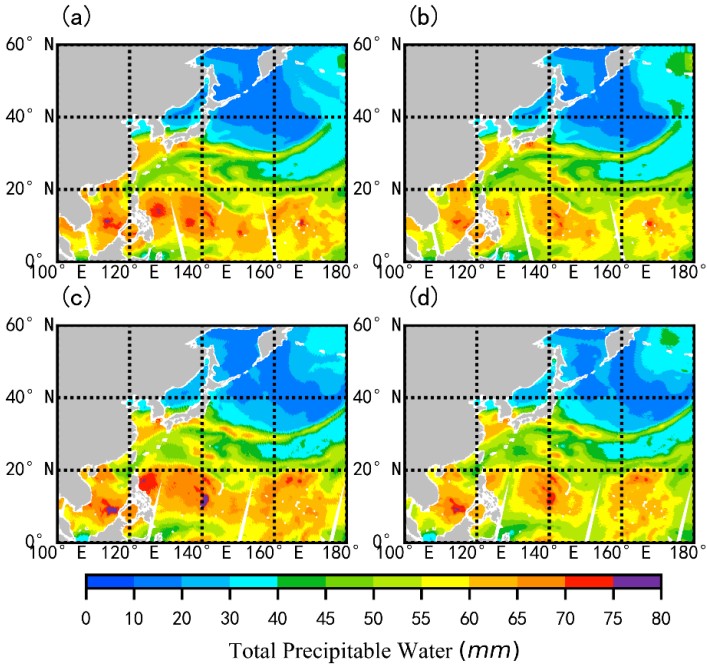

**Figure 10.** *TPW* in the northwest Pacific on 13 September, 2018 retrieved from ATMS by using (**a**) the statistical algorithm and (**b**) the physical algorithms from the ascending orbit; (**c,d**) those from the descending orbit.

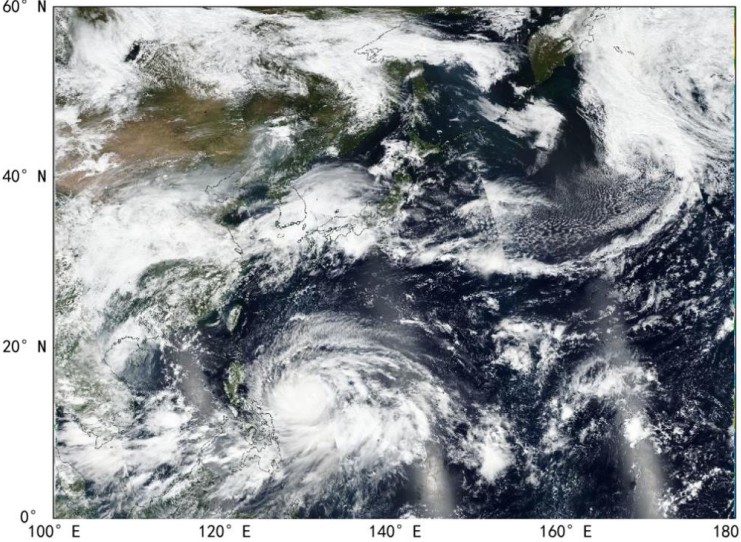

**Figure 11.** VIIRS visible band image in northwest Pacific on 13 September 2018.

Figure 12 depicts the *CLW* obtained with the MWTS-III and ATMS physical algorithms (Figure 12a,c), as well as the reanalysis data after matching with the MWTS-III (Figure 12b). To begin, a comparison of Figure 12a,b shows that the distribution of *CLW* in the global ocean obtained by the MWTS-III physical method agrees well with the ERA5 reanalysis data. The cloud system in the tropical convergence zone near the equator is clearly visible, and the positions and intensities of cloud clusters in the typhoon area in the northwestern Pacific Ocean are nearly identical in both images. Second, comparing Figure 12a,c, it can be seen that the *CLW* derived from MWTS-III and ATMS based on physical methods, though

the observation times differ, have very similar cloud system structures. This demonstrates that, in terms of liquid water, the performance of MWTS-III can be equivalent to ATMS. Because MWTS-III has a wider scan width, it can achieve global coverage, though there is still a gap between each ATMS track. With the support of new instruments, we can conduct *CLW* monitoring at various times, which is extremely beneficial to cloud detection and numerical prediction models.

Figure 13 shows the TPW of physics-based MWTS-III retrieval, the ERA5 reanalysis data after MWTS-III matching, and the ATMS retrieval result. Figure 13a,b show how the *TPW* obtained by the physical algorithm and the ERA5 reanalysis data, respectively, are distributed in the global ocean, including their intensity and range, based on MWTS-III. The only difference is a discontinuity in the *TPW* at the MWTS-III orbital junction, which is especially noticeable in the Pacific Northwest. Despite using ATMS with different observation times to retrieve TPW (Figure 13c), the results still match the first two and can even fill the "gap" at the MWTS-III orbit junction.

We compared the data from one week in July 2021 to the ERA5 reanalysis to see if the precipitable water retrieval based on the physical algorithm of the microwave sounder was accurate. Simultaneously, to reduce the retrieval error caused by the large-angle observation, we filtered the results of the two scan lines before and after MWTS-III and ATMS, respectively, and compared them to the ERA5 data using simultaneous space matching. The results are shown in Figure 14. The good news is that, for both sounders, the *TPW* retrieved by the physical algorithm is highly correlated with the ERA5 reanalysis data.

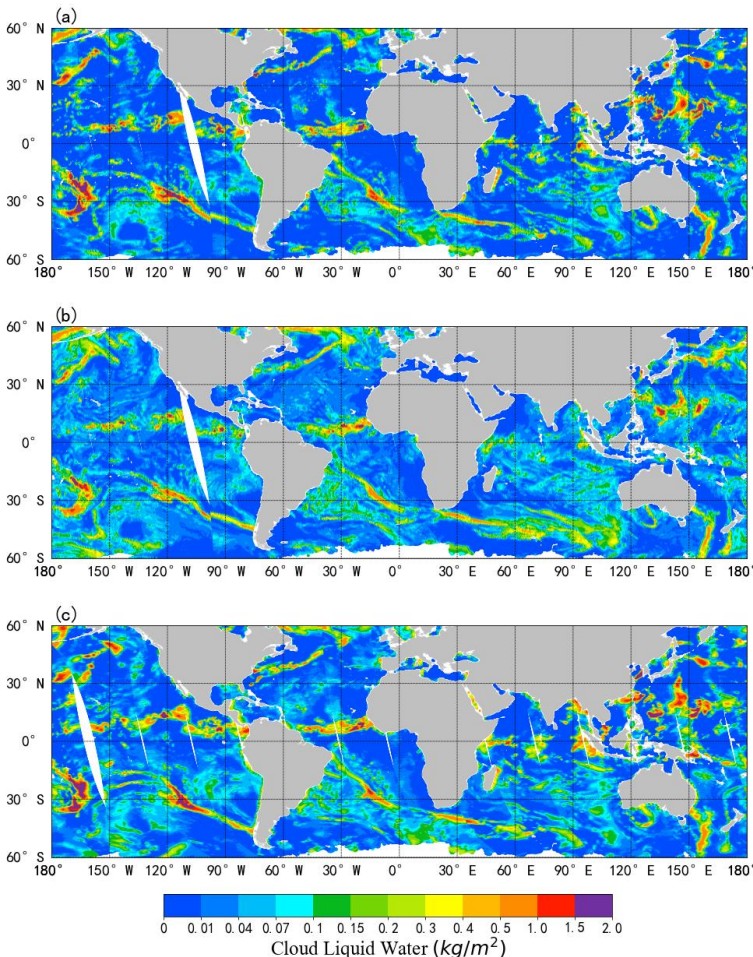

**Figure 12.** *CLW* retrieved from MWTS-III and ATMS over global oceans on 1 August 2021. (**a,c**) The cloud liquid water path retrieved from of MWTS-III and ATMS using the physical algorithm; (**b**) *CLW* from ERA5 reanalysis at the time of MWTS-III.

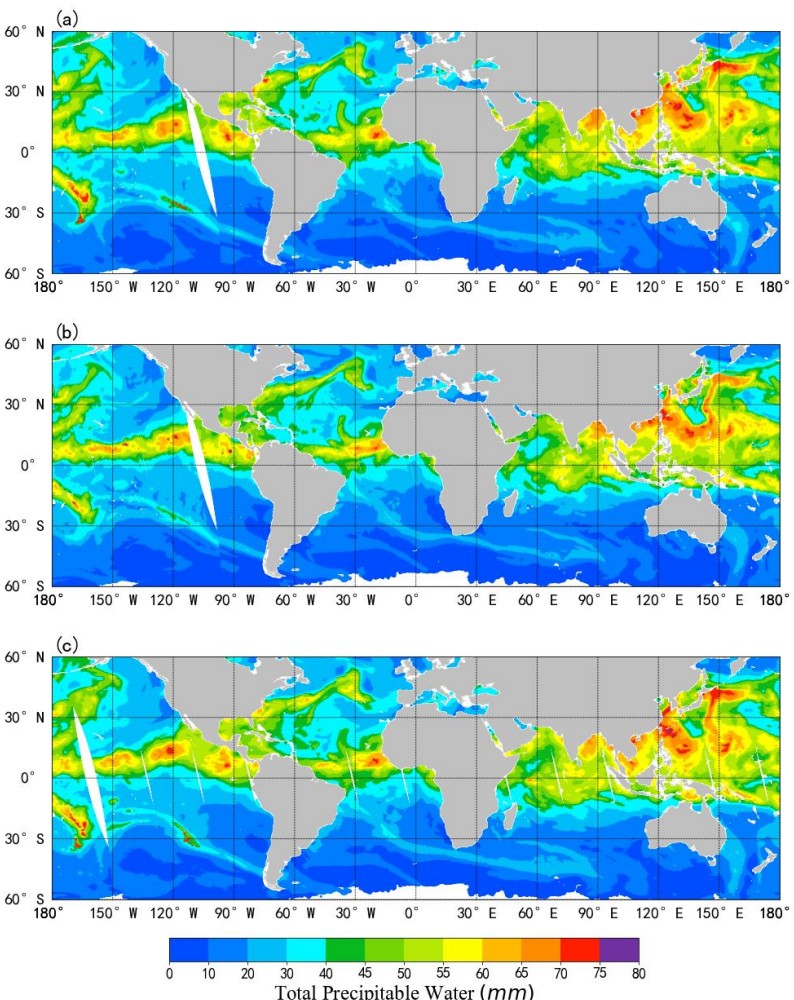

**Figure 13.** *TPW* retrieved from MWTS-III and ATMS over global oceans on 1 August 2021. (**a**,**c**) The cloud liquid water path retrieved from of MWTS-III and ATMS using the physical algorithm; (**b**) *TPW* from ERA5 reanalysis at the time of MWTS-III.

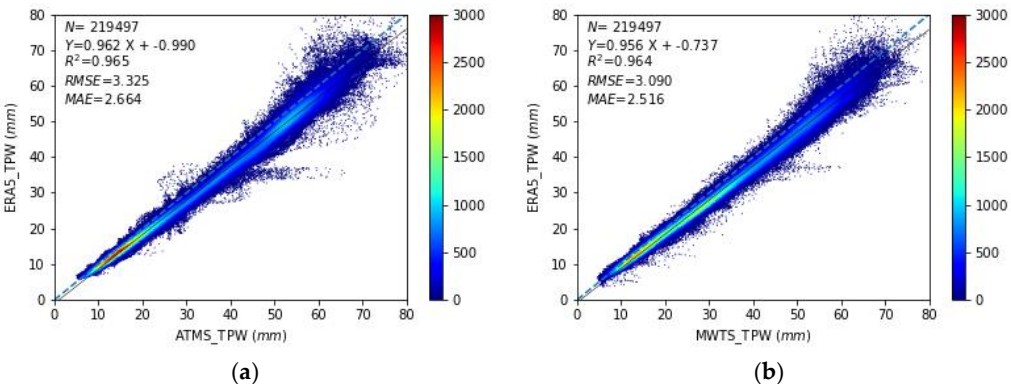

**Figure 14.** Scatter plot of *TPW* over oceans from (**a**) ATMS and (**b**) MWTS-III ascending orbit data collocated with ERA5 *TPW* on 1 August 2021.

## 5. Summary and Conclusions

In this study, the polarization of the first two MWTS-III channels are determined through radiative transfer simulations. An asymmetric correction is applied to correct the observed brightness temperatures to the values computed from the emission-based radiative transfer model. With ATMS observations, the *CLW* and *TPW* of different algorithms

are compared and analyzed. The same physical algorithm is then applied to MWTS-III and can yield the *CLW* and *TPW* from the early-morning orbit satellites. Both *CLW* and *TPW* are also compared to ERA5 reanalysis data. It is demonstrated that the global distribution of MWTS and ATMS cloud water are fairly consistent. MWTS-III can detect the clouds without the orbital gaps due to its wide scan swath. For *TPW* retrievals, when compared to ERA reanalysis, MWTS-III has product accuracy comparable to ATMS.

It should be highlighted that the physical retrieval requires some input parameters, such as sea surface temperature, surface wind, and cloud temperatures. The current algorithm can be easily implemented in a numerical weather prediction model (NWP) for a quality control of satellite data assimilation. For a general remote sensing application, the knowledge of these parameters is another area of research. More ATMS and MWTS-III sounding channels are being used to develop novel comprehensive algorithms for retrieving cloud liquid water and water vapor profiles, as well as surface parameters. The uncertainty in surface parameters on the retrievals of CLW and TPW is also subject to our further research.

**Author Contributions:** Conceptualization, C.D. and F.W.; methodology, C.D. and F.W.; validation, C.D. and J.Y.; writing—original draft preparation, C.D.; writing—review and editing, F.W. All authors have read and agreed to the published version of the manuscript.

**Funding:** This research was funded by the National Key Research and Development Program of China (2021YFB3900400), the National Natural Science Foundation of China (U2142212).

**Data Availability Statement:** The ATMS data can be downloaded from the website of NOAA Comprehensive Large Array-data Stewardship System (CLASS) (https://www.avl.class.noaa.gov/saa/products/welcome, accessed on 26 November 2021). The ERA5 reanalysis dataset can be downloaded from the website of the ECMWF Climate Data Store (https://cds.climate.copernicus.eu/cdsapp#!/home, accessed on 26 November 2021).

**Acknowledgments:** We thank Hao Hu, Yang Han, and Yining Shi for guiding the use of ARMS radiative transfer model. We also sincerely thank the reviewers and the editors for their insightful comments.

**Conflicts of Interest:** The authors declare no conflict of interest.

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
