# Peer review of "Assessments of Cloud Liquid Water and Total Precipitable Water Derived from FY-3E MWTS-III and NOAA-20 ATMS"

_remotesensing, doi:10.3390/rs14081853_

Round 1

Reviewer 1 Report

The manuscript contains major flows and is not suitable for publication in its present form. The most critical elements are:

1) The authors rely heavily on self-citation with research that goes back to 2003 and ignored most of the recent literature on the retrieval of TPW and CLW from passive microwave.

2) The CLW and TPW derived from ATMS and MWTS-III are evaluated against the ERA-5 reanalysis. However, the ERA-5 reanalysis assimilates the ATMS radiances as well as the radiances of several similar microwave instruments. This critical element is not mentioned anywhere in the manuscript.

3) The manuscript contains a discussion about “O”, “B” and “O-B”. These quantities are not defined anywhere and, to be honest, I have absolutely no idea what they are.

4) The authors claim that, because on one single case study the correlation between MWTS-III TPW and ERA-5 TPW is 0.971 against 0.969 between ATMS TPW and ERA-5 TPW, “MWTS-III has a higher correlation with ERA reanalysis than ATMS”. The least I can say is that I am not convinced that this “improvement” is statistically significant (note that, because of the spatial correlation of TPW, each pixel cannot be considered as an independent sample when evaluating the significance).

Additional comments are made directly on the PDF file.

Author Response

Dear reviewer,

Thank you so much for your feedback,we have revised the content of the article according to your comments.

Reviewer 2 Report

Review of the paper:

Assessments of cloud liquid water and total precipitable water derived from FY-3E MWTS-III and NOAA-20 ATMS

By: Changjiao Dong, Fuzhong Weng, and Jun Yang

General comment

This paper shows the computation Liquid Cloud liquid water (CLW) and Total Precipitable Water (TPW) from Advanced Technology Microwave Sounder (ATMS) and Microwave Temperature Sounder (MWTS-III), the former carried onboard the National Polar-orbiting Partnership (Suomi NPP) satellite of the United States' new-generation Joint Polar Satellite System (JPSS), and the second onboard Fengyun-3E (FY-3E) early morning satellite.

The paper also gives a method to evaluate the polarization of the MWTS-III through radiative transfer calculations. It is found that the CLW and TPW derived from two instruments exhibit a high consistency in terms of their spatial distributions and magnitudes and that they are consistent with ERA5 reanalysis and VIIRS visible band image.

The paper is, in general, well written and interesting. I have, however, some suggestions that could improve the quality of the paper.

Major points

I suggest revising your paper according to the following points:

  1. The difference between the statistical and physical methods is not clearly stated. Few sentences must be dedicated to better explain the difference in Section 3 or at the start of Section 4.
  2. Comparison with ERA5: while I can understand the need to use ERA5 data for verification/comparison, they are analyses and not direct observations of CLW and TPW. Could the author suggest/discuss some observed dataset for comparison?

Minor points

The paper is well written and consistent. I found few minor points and they are reported in the sticky notes attached to the pdf of the paper.

Author Response

(The authors gave the same response as above.)

Round 2

Reviewer 2 Report

The paper was revised according to my comments. The paper can be accepted as is. Regards.